# Echocardiographic and Cardiac MRI Comparison of Longitudinal Strain and Strain Rate in Cancer Patients Treated with Immune Checkpoint Inhibitors

**DOI:** 10.3390/jpm12081332

**Published:** 2022-08-19

**Authors:** Jibran Mirza, Sunitha Shyam Sunder, Badri Karthikeyan, Sharma Kattel, Saraswati Pokharel, Brian Quigley, Umesh C. Sharma

**Affiliations:** 1Department of Medicine, Division of Cardiology, Jacob’s School of Medicine and Biomedical Sciences, Buffalo, NY 14203, USA; 2Department of Pathology and Laboratory Medicine, Roswell Park Comprehensive Cancer Center, Buffalo, NY 14203, USA; 3Clinical and Research Institute on Addictions, University at Buffalo, Buffalo, NY 14203, USA

**Keywords:** cardiac MRI, cardiotoxicity, checkpoint inhibitors, echocardiography, immunotherapy, strain imaging

## Abstract

Background: Immune checkpoint inhibitor (ICI)-induced cardiac side effects in cancer patients are increasingly being recognized and can be fatal. There is no standardized cardiac imaging test to examine the effects of ICIs in myocardial morphology and function. Objective: To study the utility of echocardiography and cardiac MRI in examining regional and global changes arising from ICI-induced myocarditis and cardiomyopathy in high-risk subjects suspected to have developed ICI cardiomyopathy. Methods: We studied eight consecutive patients referred for cardiac MRI (CMR) from a comprehensive cancer center for suspected ICI-induced myocarditis and compared the data with sixteen age-matched controls. Using newly developed strain analysis algorithms, we measured myocardial strain and strain rates using echocardiography and CMR. Then, we compared the mean longitudinal strain and strain rates derived from echocardiography and CMR in the same ICI-treated cohort of patients (*n* = 8). They underwent both of these imaging studies with images taken 24–48 h apart and followed up prospectively within the same hospital course. Results: All our cases had preserved ejection fraction (EF) > 50%. Echocardiogram showed reduced mean systolic longitudinal strain (LS, %) (ICI: −12.381 ± 4.161; control: −19.761 ± 1.925; *p* < 0.001), peak systolic strain rate (SR_S_, s^−1^) (ICI: −0.597 ± 0.218; control: −0.947 ± 0.135; *p* = 0.002) and early diastolic strain rate (SR_E_, s^−1^) (ICI: 0.562 ± 0.295; control: 1.073 ± 0.228; *p* = 0.002) in ICI-treated cases. Direct comparison between the echocardiogram vs. CMR obtained within the same hospital course demonstrated strong a correlation of LS scores (*r* = 0.83, *p* = 0.012) and SR_S_ scores (r = 0.71, *p* = 0.048). The Bland–Altman plots showed that 95% of the data points fitted within the ±1.96 SD of the mean difference, suggesting an agreement among these two imaging modalities. Conclusion: In this feasibility cohort study, both echocardiography- and CMR-based strain indices illustrate changes in myocardial contractility and relaxation suggestive of ICI-induced cardiomyopathy. Our data, after validation in a larger cohort, can form the basis of myocardial imaging in cancer patients treated with ICIs.

## 1. Introduction

Advances in cancer treatment have grown to include a new subset of therapy known as immune checkpoint inhibitors (ICIs), which are at the forefront of antineoplastic therapies for several types of cancers. These include anti-cytotoxic T lymphocyte-associated antigen CTLA-4 (e.g., ipilimumab, tremelimumab), anti-PD-1 (nivolumab, pembrolizumab, cemiplimab) and anti-PD-L1 (atezolizumab, durvalumab, avelumab). ICI therapy inhibits the immune evasion of tumor cells, allowing for T cell-mediated antitumor immunity [1].

While ICI therapies have shown promising results, their toxicity profile includes various autoimmune and inflammatory conditions including cardiotoxicity [2,3,4,5,6,7,8,9,10,11,12,13,14]. Cardiotoxicity profile includes systolic heart failure, arrhythmias, pericardial and myocardial disease of which the most common reported is myocarditis when utilized to treat melanoma [15,16]. ICI-induced myocarditis is associated with a high mortality rate of 46% [17], with a prevalence of 0.06–2.4% [12] and incidence of about 1%, which doubles with combination therapy [17,18]. Presentation is extremely variable, ranging from asymptomatic cardiac biomarker elevation to life-threatening fulminant myocarditis [19,20]. Screening begins with serial troponin measurements and electrocardiography (ECG), though there are no clear evidence-based guidelines [18,19,21,22]. A 10 ms increment in QRS duration in post-ICI myocarditis ECG is associated with increased odds of MACE [23,24]. Non-invasive imaging modalities such as echocardiogram and cardiac magnetic resonance (CMR) allow for better tissue characterization. Endomyocardial biopsy (EMB) is the gold standard for confirmation; however, at least six biopsies from different regions are needed given the patchy T cell myocardial infiltration, even so there is a high false negative rate [12]. A retrospective study conducted CMR strain analysis, indicating that despite normal left ventricular function, strain imaging was abnormal, thus highlighting its importance [25].

In this study, we aim to further study the utility of CMR in comparison to echocardiography in patients suspected to have developed ICI-related adverse events. In particular, we focused on the regional and global comparisons of mean longitudinal systolic strain, peak systolic strain rate and early diastolic strain rate between these modalities.

## 2. Methods

This is a multimodality analytical study in a select group of patients selected from a single, tertiary care center, Buffalo General Hospital (BGH) and Gates Vascular Institute (GVI) in Buffalo, New York. The study was approved by the University at Buffalo Institutional Review Board. An institutional database of CMR’s from January 2017 through March 2020 were reviewed to identify patients under clinical suspicion for ICI myocarditis who underwent CMR. Age-matched population-based controls were obtained from the same database. As a result, a total of 8 cases and 16 controls were gathered to analyze echocardiography and contrast-enhanced CMR, and they were followed up prospectively with images taken 24–48 h apart.

### 2.1. Definitions and Outcomes of Interest

Myocarditis was diagnosed by several criterion including clinical suspicion, troponin elevation with/without symptoms, ECG changes and functional and/or structural abnormalities as noticed on echocardiography and CMR [19]. Patients were categorized in terms of their severity of myocarditis through using the Common Toxicity Criteria for Adverse Events (CTCAE) presented by the American Society of Clinical Oncology. With these criteria, cases were regarded as “severe acute myocarditis” or “subacute myocarditis” depending on if they were Grade 3–5 according to CTCAE or Grade 2 or less according to CTCAE, respectively. MACE was defined by a composite of cardiogenic shock, cardiac arrest, complete heart block (CHB) and cardiac death.

### 2.2. Covariates

The electronic medical records were accessed to obtain patient demographics, pertinent cardiac and medical history, electrocardiogram, echocardiography and cardiac biomarkers (troponin, BNP, CK, CKMB, Myoglobin). Patients’ clinical presentation on initial evaluation, initial and peak troponin values and any additional pertinent cardiac imaging to evaluate etiology of presenting symptoms were included. In terms of cancer specific co-variates, cancer type, ICI treatment, single vs. combination ICI and doses of ICIs were obtained. Reports of radiotherapy and chemotherapy were also recorded.

### 2.3. Strain and Strain Rate Analysis

All cases (*n* = 8) and controls (*n* = 16) underwent analysis of cardiomyocyte contraction using principles of finite strain theory, as described previously [26,27]. Briefly, one-dimensional Lagrangian strain (*ε*) was calculated along the longitudinal directions. Lagrangian strain is defined as the change in length of the myocardial segment (*L*) from the original length (*L*_0_) at end diastole, as shown below [28].
$$\varepsilon =\frac{L-L_{0}}{L_{0}}$$

Similarly, cardiomyocyte contraction and relaxation were conducted utilizing Eulerian strain rates (*SR*). Eulerian strain rate, which is based on the myocardial velocity gradient (derived from velocities *v*_1_ and *v*_2_), measures the change in strain with respect to the time at peak systole (*SR*_S_, s^−1^) and early diastole (*SR*_E_, s^−1^), as described by the following formula [28].
$$SR=\frac{v_{2}-v_{1}}{L}=\frac{1}{L}\frac{dL}{dt}=\frac{1}{\varepsilon +1}\frac{d\varepsilon }{dt}$$

Longitudinal strain was calculated under two-chamber, three-chamber and four-chamber long-axis images. Left ventricular (LV) and right ventricular (RV) borders were manually drawn at end diastole and the software outlined the remainder of the endocardial border. Any further adjustment of the initial borders was manually performed by an experienced technician/qualified imaging professional. LV strain and strain rates were calculated globally, as well as in basal, midventricular and apical regions. RV strain and strain rates for the free wall and septal regions were also measured to analyze RV contractility.

### 2.4. Echocardiography Protocol for Strain Evaluation

Echocardiography images were downloaded in Digital Images and Communications in Medicine (DICOM 3, DICOM^®^, Arlington, VA) and transmitted to an experienced imaging interpreter. Studies were uploaded and studied offline using the vendor independent TomTec software module (TOMTEC USA, Chicago, IL, USA). Two-dimensional speckle tracking echocardiography was carried out using TomTec AutoSTRAIN, Version; Image-Com5 5.5.4.467461 to calculate segmental and longitudinal strain of the visualized left ventricle segments, as described previously [27]. The endocardium was visualized and demarcated during end systole and diastole. The area of interest was automatically traced by the software and the magnitude of deformation was used to generate strain curves.

### 2.5. Contrast-Enhanced CMR Protocol

Imaging of selected patients was performed with a GE 1.5-T scanner with manufacturer recommended technical parameters. Images were obtained with patients in the supine position, with an inspiratory breath-hold and ECG gating. Scout images were taken in coronal, sagittal and axial planes. CMR protocol included balanced cine steady-state free precession (SSFP) imaging for cardiac function and mass, T1-weighted fast spin echo (FSE) sequence before and after intravenous (IV) gadolinium injection and T2-weighted triple-inversion recovery images. Delayed enhanced images were taken within 10 min of obtaining contrast. Segment version 3.2 R8531 (http://segment.heiberg.se, accessed on 21 July 2022) was utilized to evaluate cardiac function and perform strain analysis, as described previously [26,29,30,31,32] LGE were categorized based on the more predominant pattern as sub-endocardial/transmural, sub-epicardial, mid-myocardial and diffuse enhancement. CMR studies were analyzed by qualified professionals at Buffalo General Hospital.

### 2.6. Statistical Analyses

Quantitative endpoints were summarized by the group using mean and standard error of the mean (SEM). To compare findings from both ICI-treated and control groups on systolic and diastolic strain data between groups, independent samples Student’s *t*-tests with a two-sided significance level set at level of 0.05 were conducted. These tests were carried out separately for echocardiography and CMR data. Because of the small sample sizes in this study, effect sizes were also calculated for all comparisons. Cohen’s d provides a standardized difference between treatment groups calculated by dividing the difference in the mean of the two groups by the pooled standard deviation between groups. This effect size indicates how great a difference in standard deviations exists between the two treatment groups’ averages. A Cohen’s d of 0.2 is considered a small effect size, 0.5 a medium effect size and 0.8 or above a large effect size. Because Cohen’s d gives a biased estimate for sample sizes smaller than 20, a correction factor is multiplied to Cohen’s d to create an unbiased Hedges’s g statistic [33]. Correlations between echocardiography and CMR findings are also presented. Inter-modality agreement between the quantitative measurements of mean longitudinal strain and strain rates calculated by using echocardiography and CMR were presented with Bland–Altman plot and correlation analysis. Graph Pad Prism 9.1.2 (La Jolla, CA, USA) software was employed for drawing the Bland–Altman diagrams to evaluate the difference and the mean. If the difference in the two tests was within the consistency limits, it was clinically acceptable and denoted as good consistency. *p*-values less than 0.05 were considered statistically significant.

The interrater reliability was measured as per Cohen’s kappa statistic. B.K and S.S.S analyzed echocardiography findings with a kappa value of 0.500 (95% CI −0.020 to 1.0000) suggestive of moderate agreement. U.C.S. and S.S.S. analyzed CMR had a kappa value of 1.0000 (95% CI 1.0000 to 1.0000) reported an almost perfect agreement. These values were calculated using Graph Pad Prism 9.1.2 (La Jolla, CA, USA) software.

## 3. Results

### 3.1. Patient Characteristics

All eight cases were included in the study. There were three females (37.5%) and five males (62.5%). The mean age was 69.1 years (Range 54–80). Comorbidities from most to least common were hypertension (*n* = 7, 87.5%), diabetes (*n* = 5, 62.5%), hypothyroidism (*n* = 4, 50%) and atrial fibrillation (*n* = 3, 37.5%). None of the cases (*n* = 0, 0%) were diagnosed with coronary artery disease or myocardial infarction. The baseline clinical characteristics are shown in Table 1.

### 3.2. Cancer Type and ICI Treatment Characteristics

Malignancy from most to least common were metastatic melanoma (*n* = 4, 50%), non-small cell lung carcinoma (*n* = 2, 25%), small cell lung carcinoma (*n* = 1, 12.5%) and peritoneal carcinomatosis (*n* = 1, 12.5%). The pattern of ICI use was as follows: pembrolizumab (*n* = 3, 37.5%), nivolumab (*n* = 3, 37.5%), ipilimumab (*n* = 2, 25%), avelumab (*n* = 1, 12.5%), durvalumab (*n* = 1, 12.5%) and combination ipilimumab and nivolumab (*n* = 1, 12.5%). From initial ICI dose and the onset of symptoms, the mean number of days was 46 days, the median number of days was 35 days and the mode was 21 days, as shown in Table 1.

### 3.3. Cardiac Biomarkers

Mean troponin at the onset was 1.79 ng/mL (range 0.24–6.18 ng/mL). Mean peak troponin value was 2.52 ng/mL (range 0.28–6.18 ng/mL). The number of cases with troponin values that had peaked by admission was six (75% of cases). The mean BNP value was 136 pg/mL (range 34–318 pg/mL). Of note, three out of eight (*n* = 3, 37.5%) cases had BNP values measured. The mean CK value was 960 IU/L (Range 160–2494 IU/L). Similarly, two out of eight (*n* = 2, 25%) cases had CKMB values measured. The mean CKMB value was 87.5 ng/mL (Range 10–165 ng/mL). In addition, six out of eight (*n* = 6, 75%) cases had myoglobin measured. The mean myoglobin level was 1220 mg/dL (Range 39–3612 mg/dL). The data for cardiac biomarkers are presented in Table 1.

### 3.4. Myocarditis Onset, Grading, Treatment and Outcomes

The most common to least common symptoms or reasons for presentation were dyspnea on exertion (*n* = 4, 50%), chest pain (*n* = 3, 37.5%), lightheadedness (*n* = 1, 12.5%), incidental troponin elevation (*n* = 1, 12.5%) and presyncope (*n* = 1, 12.5%). All cases were found to have severe myocarditis criteria by CTCAE Myocarditis Grading. Treatment was primarily steroids in 87.5% of cases, and one case (12.5%) was treated empirically for coronary artery disease. Four cases (50%) were started on steroids on day 0, one case (12.5%) on day 1, one case (12.5%) on day 3 and one case (12.5%) on day 21 of presentation. In addition, one patient received monoclonal antibody infliximab (12.5%) in addition to steroid therapy. Five cases (62.5%) are alive and three cases (37.5%) are deceased. Cause of death includes one case with atrial fibrillation and rapid ventricular rate inducing congestive heart failure (CHF) (12.5%), one case with a cerebral vascular accident (CVA) (12.5%) and one case with multiorgan failure from metastatic melanoma (12.5%). The clinical characteristics are summarized in Table 2.

### 3.5. Conduction Abnormalities in Electrocardiogram

New right bundle branch block (RBBB) was present in one (12.5%) case. Other changes included sinus bradycardia (*n* = 2, 25%), sinus bradycardia with first degree atrioventricular (AV) block (*n* = 1, 12.5%), sinus rhythm with old RBBB (*n* = 2, 25%), normal sinus rhythm (*n* = 3, 37.5%), sinus tachycardia (*n* = 2, 25%) and old atrial flutter (*n* = 1, 12.5%). All cases had echocardiograms with preserved ejection fraction (EF). One case (12.5%) had stage I diastolic dysfunction (Table 3).

### 3.6. Echocardiographic Characteristics

A representative echocardiogram of an ICI-treated patient is shown in Figure 1. Mean longitudinal strain (LS), peak systolic strain rate (SR_S_) and early diastolic strain rate (SR_E_) of healthy controls and ICI-treated cases were analyzed. Representative strain curves illustrating changes in strain over time are shown in Figure 2. Strain and strain rate characteristics for regional and global longitudinal contractility are presented in Table 4 and Table 5. As shown in Table 5, LS, SR_S_ and SR_E_ were all significantly lower in the basal and midventricular regions in ICI-treated patients compared to the controls. While the apical region experienced a trend in LS decline (*p* = 0.054) in ICI-treated patients, there was no statistically significant difference in SR_S_ or SR_E_ in the apical region, which might be suggestive of apical sparing in ICI-treated patients.

### 3.7. Cardiac MRI Characteristics

The same cohort of eight ICI-treated patients who underwent echocardiogram analyses were also examined by cardiac MRI and were compared with eight CMR controls. Abnormal LGE patterns were noted on 37.5% (*n* = 3) of our patients on presentation through CMR. Figure 3 illustrates CMR-based longitudinal strain changes in a single ICI-treated case vs. control. CMR-derived left ventricular (LV) longitudinal strain and strain rate measurements for global and regional myocardial function are presented in Table 6 and Table 7. As seen in Table 7, while there is no statistically significant difference in LV longitudinal strain between ICI-treated patients and controls, the effect size of *g* = 0.803 suggests the true effect might be large and the lack of significant differences could be due to small sample size and low statistical power. However, ICI-treated patients did experience statistically significant global decreases in LV SR_S_ and SR_E_ compared to the controls. Additionally, cardiac MRI seemed to suggest apical sparing in ICI-treated patients, which was also observed by echocardiography.

As illustrated in Table 8, ICI-treated patients had significantly reduced right ventricular (RV) longitudinal strains both globally as well as in the free wall and septal regions. Compared to the controls, ICI-treated patients experienced significant reductions in RV SRs in the septal region and RV SR_E_ in the free wall region. These results indicated that ICI-associated cardiotoxicity might have an adverse impact on RV contractility.

### 3.8. Comparison of Echocardiography vs. Cardiac Magnetic Resonance Imaging-Based Strains

Mean longitudinal strain and strain rates derived from echocardiography and CMR were compared in the same ICI-treated cohort of patients (*n* = 8) who underwent cardiac imaging studies using both of these modalities. Echocardiography and CMR LS scores were correlated, *r* = 0.83, *p* = 0.012, as were SR_S_ scores, r = 0.71, *p* = 0.048. The correlation between echocardiography and CMR SR_E_ did not reach statistical significance, *r* = 0.42, *p* = 0.301, due to the small sample size. The correlation analysis carried out for echocardiography- versus CMR-derived mean longitudinal strain and strain rates showed moderate correlation (R2 values of mean longitudinal systolic strain, mean longitudinal peak systolic strain rate and mean longitudinal early diastolic strain rate are 0.68, 0.50 and 0.02, respectively) (Figure 3C). The Bland–Altman plots showed that 95% of the data points lie within the ±1.96 SD of the mean difference—limits of agreement (Figure 3D). This suggests an agreement among the two techniques.

## 4. Discussion

Clinically, cardiac biomarkers and imaging modalities, particularly echocardiography and CMR, are used in the diagnosis of ICI myocarditis [34], but there are limited data comparing the clinical utility of newly developed strain-based imaging algorithms. The unique aspect of our study is the head-to-head comparison of echocardiography and CMR strain indices to assess any similarities or discrepancies that exists. Literature has shown that LS decreases in ICI myocarditis patients, with preserved or reduced EF compared to those without myocarditis [24]. Our study expands on the prior evidence by serially comparing cardiac morphology and function. Although cancer patients without ICI therapy would also serve as controls for such studies, it is uncommon to develop abnormal cardiac functional indices or perform cardiac imaging studies just because of the presence of cancers.

Previously, CMR LGE patterns were compared to EMB-proven ICI myocarditis, and the results indicated that less than half (40%) of these patients had corresponding LGE findings, thus recommending caution when used. EMB was not performed on our patients; however, 37.5% (*n* = 3) of our patients presented with abnormal LGE patterns on CMR, similar to the findings reported by Zhang et al. [17]. Current literature has only one study analyzing strain indices on CMR [25]. Our study thus adds to the existing literature by elucidating the utility of these diagnostic modalities.

Cardiac biomarkers, in particular Troponin (I), are the initial screening test for myocardial injury and ICI toxicity [35]. Electrocardiography (ECG) lacks sensitivity and specificity for diagnosis [36]. Our study indicates that ECG findings remained relatively unchanged from baseline (Table 3). Of the eight study patients, one patient presented with new onset RBBB, another with new onset atrial tachycardia/flutter with ST depressions and no new rhythm changes identified in the others with a 37.5% mortality. Thus, early diagnosis is paramount for the prompt initiation of therapy [37]. CMR was conducted within 4 days of diagnosis, with only three abnormal LGE patterns were observed in the patients.

## 5. Study Limitations

The limitations of our study include challenges associated with imaging in patients with permanent pacemakers and implantable cardiac defibrillators. Gadolinium contrast, although relatively safe, must be used in caution in patients with a GFR < 30, with risk for nephrogenic systemic fibrosis. The MRI machine environment can lead to claustrophobia and issues related to breath holding and imaging compliance [38]. Few limitations can be overcome with a larger sample size and potentially different study designs. Our study was a cohort study with a small sample size and both echocardiographic and CMR parameters were retrospectively analyzed. However, the calculated effect sizes are predominantly large, which suggests this study is an important step for the development of larger and more comprehensive studies examining ICI-associated cardiotoxicity. Adding baseline measurements, potentially baseline echocardiography and CMR prior to beginning ICI treatment, may significantly contribute to the clinical implications of these data. However, current guidelines do not recommend cardiac imaging in subjects anticipating ICI administration. Furthermore, there are no data on long-term follow-up patients who have been successfully treated for ICI myocarditis. For non-ICI myocarditis, data have indicated that at 6 months, LGE without edema could have worse prognosis and possibly fibrosis when located at the mid-septal wall, whereas with edema, residual recovery is possible [39]. Thus, repeat follow up with CMR to identify prognosis should be studied.

## 6. Conclusions

In this multimodality retrospective feasibility study, both echocardiography- and CMR-derived strain indices illustrate changes in myocardial contractility and relaxation suggestive of ICI-associated cardiomyopathy. Our data, after validation in a larger cohort, can form the basis of myocardial imaging in cancer patients treated with ICIs.

## Figures and Tables

**Figure 1 jpm-12-01332-f001:**
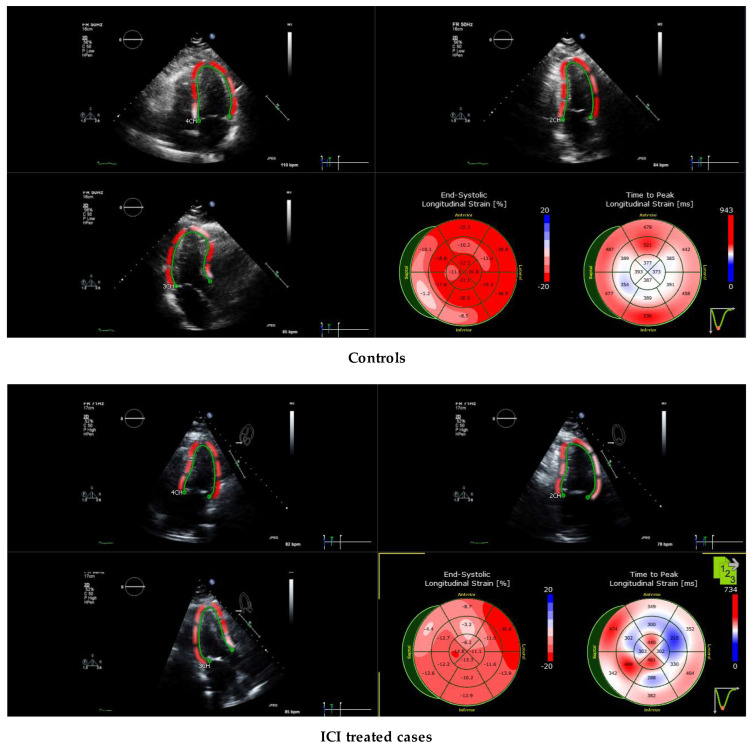
Left ventricular (LV) longitudinal strain imaging of immune checkpoint inhibitor (ICI)-induced cardiomyopathy using speckle tracking echocardiography. The midventricular and apical segments are relatively hypokinetic.

**Figure 2 jpm-12-01332-f002:**
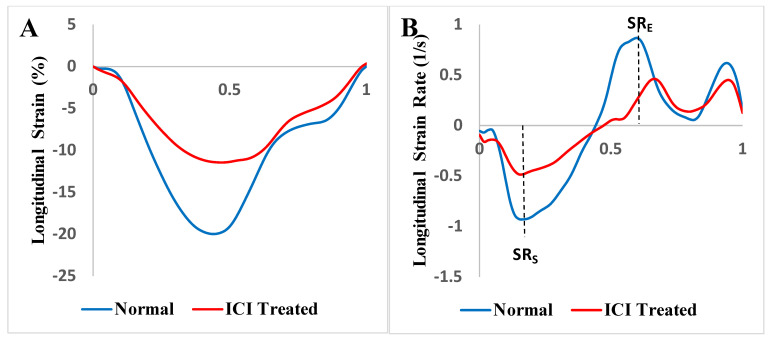
Strain curves illustrating changes in left ventricular (LV) longitudinal strain and strain rate over the course of one cardiac cycle. Strain curves were generated by averaging individual strain curves calculated at each time point. *n* = 8 normal and immune checkpoint inhibitor (ICI)-treated patients. A. LV, longitudinal strain curves; B. LV, longitudinal strain rate curves. SR_S_ = peak systolic strain rate; SR_E_ = early diastolic strain rate.

**Figure 3 jpm-12-01332-f003:**
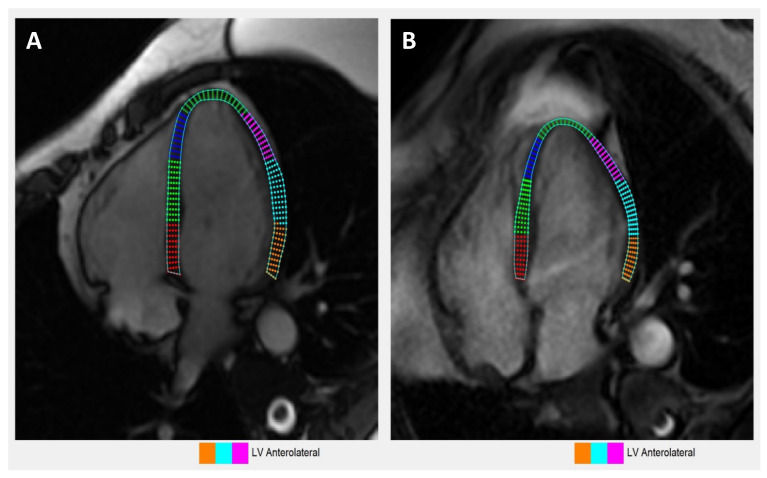
Left ventricular (LV) longitudinal strain imaging of immune checkpoint inhibitor (ICI)-induced cardiomyopathy using cardiac MRI (CMR) under 4 chamber view (4CH). (**A**) Representative 4CH-CMR image of a control; (**B**) Representative 4CH-CMR image of an ICI-treated patient. (**C**) Scatter plots showing the correlation between the mean longitudinal strain and strain rates derived from echocardiography versus CMR. (**D**) Bland–Altman agreement plots for ICI-induced cardiomyopathy detected on echocardiography compared with CMR for mean longitudinal strain and strain rates. Bland–Altman plots display the difference between values measured by echocardiography and CMR against the mean of these 2 values. The upper and lower dashed lines represent the 95% confidence intervals.

**Table 1 jpm-12-01332-t001:** Demographic characteristics of patients, cancer type, immune checkpoint inhibitor (ICI) treatment and cardiac biomarkers.

Baseline Characteristics	
Age, years	69.1 (54–80)
Female, %	37.5
Male, %	62.5
Risk factors and past medical history	
Atrial fibrillation, %	37.5
Hypertension, %	87.5
Hyperlipidemia, %	12.5
Hypothyroidism, %	50.0
Diabetes mellitus, %	62.5
Dementia, %	12.5
Pulmonary embolism, %	12.5
Cancer	
Metastatic melanoma, %	50%
Non-small cell lung carcinoma, %	25%
Small cell lung carcinomatosis, %	12.5%
Peritoneal carcinomatosis, %	12.5%
Immune checkpoint inhibitor	
Pembrolizumab, %	37.5%
Nivolumab, %	37.5%
Avelumab, %	12.5%
Ipilimumab, %	25%
Durvalumab, %	12.5%
Combination Ipilimumab and Nivolumab, %	12.5%
Average days from first dose and onset of toxicity from ICI initiation	
Mean, days	46
Median, days	35
Mode, days	21
Cardiac biomarker data	
Mean Troponin at onset of symptoms, ng/mL	1.79 (Range 0.24–6.18)
Mean Troponin Peak, ng/mL	2.52 (Range 0.28–6.18)
Mean Maximum BNP, ng/mL	136 (Range 34–318 pg/mL)
Mean Maximum CK-MB, ng/mL	960 (Range 160–2494)
Mean Maximum Myoglobin, mg/dL	1220 (39–3612)

BNP, brain natriuretic peptide; CK-MB, creatinine kinase MB.

**Table 2 jpm-12-01332-t002:** Immune checkpoint inhibitor (ICI)-associated myocarditis clinical presentation, cardiac biomarkers, treatment regimen and major adverse cardiac events (MACE).

Signs and Symptoms	
Cerebral vascular event	1 case(s)
Chest pain	3 case(s)
Dizziness	1 case(s)
Dyspnea on exertion	4 case(s)
Incidental troponin elevation	1 case(s)
Lightheadedness	1 case(s)
Orthopnea	1 case(s)
Paroxysmal nocturnal dyspnea	1 case(s)
Palpitations	1 case(s)
Pruritis	1 case(s)
Side effects beyond myocarditis	
Hepatitis	2 case(s)
None	4 case(s)
Thyroiditis	1 case(s)
Vision changes	1 case(s)
CTCAE grading for myocarditis	
Grade 3	7 case(s)
Grade 4	1 case(s)
Steroid start time and onset of symptoms (days)	
0	4 case(s)
1	1 case(s)
3	1 case(s)
21	1 case(s)
Treatment	
Aspirin 81 mg	2 case(s)
Colchicine	1 case(s)
Rosuvastatin	1 case(s)
Ibuprofen	1 case(s)
Infliximab	1 case(s)
Methyl prednisone	5 case(s)
Prednisone	1 case(s)
Treatment outcome	
Myocarditis resolved	7 case(s)
Death from other cause	1 case(s)
Cause of death	
Cerebral vascular accident	1 case(s)
Multi-organ failure	1 case(s)
Myocardial infarction and cardiogenic shock	1 case(s)

CTCAE: Common Terminology Criteria for Adverse Events.

**Table 3 jpm-12-01332-t003:** Electrocardiography and echocardiography parameters for immune checkpoint inhibitor (ICI)-treated patients at the time of presentation with suspected cardiotoxicity.

Electrocardiography on Presentation	
Sinus bradycardia	2 case(s)
Sinus bradycardia with first degree AV block	1 case(s)
Normal sinus rhythm with right bundle branch block	1 case(s)
Sinus tachycardia with right bundle branch block	1 case(s)
Normal sinus rhythm with right bundle branch block	1 case(s)
Normal sinus rhythm	1 case(s)
Sinus tachycardia	1 case(s)
Atrial tachycardia with worsening ST depressions in inferior and precordial leads compared to previous electrocardiography	1 case(s)
Electrocardiography changes from baseline	1 case(s)
Echocardiography on presentation	
Ejection fraction ≥ 60%	4 case(s)
Ejection fraction ≥ 50%	2 case(s)
Pericardial effusion	1 case(s)
Systolic dysfunction	0 case(s)
Diastolic dysfunction	1 case(s)

AV, Atrioventricular.

**Table 4 jpm-12-01332-t004:** Echocardiographic strain indices of cases vs. controls.

Controls	Mean Longitudinal Systolic Strain (%)	Mean Longitudinal Peak Systolic Strain Rate (s^−1^)	Mean Longitudinal Early Diastolic Strain Rate (s^−1^)
1	−23.179	−1.192	1.430
2	−18.125	−0.773	0.687
3	−18.274	−0.887	1.035
4	−18.671	−0.873	0.873
5	−22.272	−1.054	1.249
6	−18.510	−0.852	1.027
7	−19.221	−0.918	1.099
8	−19.836	−1.031	1.183
Average	−19.761	−0.947	1.073
Standard Deviation	1.925	0.135	0.228
**Cases**	**Mean Longitudinal Systolic Strain (%)**	**Mean Longitudinal Peak Systolic Strain Rate (s^−1^)**	**Mean Longitudinal Early Diastolic Strain Rate (s^−1^)**
1	−17.303	−0.973	0.757
2	−11.340	−0.642	0.429
3	−11.856	−0.463	0.244
4	−10.884	−0.449	0.508
5	−18.406	−0.790	1.191
6	−12.553	−0.657	0.434
7	−11.864	−0.519	0.561
8	−4.846	−0.279	0.375
Average	−12.381	−0.597	0.562
Standard Deviation	4.161	0.218	0.295
*p* value (Controls vs. Cases)	<0.001	0.002	0.002

**Table 5 jpm-12-01332-t005:** Left ventricular (LV) strain and strain rate characteristics of immune checkpoint inhibitor (ICI)-treated patients compared to the controls using echocardiography.

Parameters	Controls (*n* = 8)	ICI-Treated Patients (*n* = 8)	*p* Value	Hedges’s g for Effect Size
LV Systolic Longitudinal Strain (%)				
Basal	−22.602 ± 5.792	−13.763 ± 2.530	0.001 (*)	1.870
Midventricular	−16.630 ± 2.292	−9.800 ± 4.126	0.001 (*)	1.935
Apical	−20.196 ± 2.877	−14.180 ± 7.564	0.054	0.994
Global	−19.761 ± 1.925	−12.381 ± 4.161	<0.001 (*)	2.152
LV Peak Systolic Longitudinal Strain Rate (s^−1^)				
Basal	−1.146 ± 0.345	−0.630 ± 0.172	0.002 (*)	1.787
Midventricular	−0.786 ± 0.135	−0.473 ± 0.134	<0.001 (*)	2.202
Apical	−0.891 ± 0.142	−0.732 ± 0.559	0.446	0.370
Global	−0.947 ± 0.135	−0.597 ± 0.218	0.002 (*)	1.830
LV Early Diastolic Longitudinal Strain Rate (s^−1^)				
Basal	1.333 ± 0.446	0.627 ± 0.210	0.001 (*)	1.913
Midventricular	0.811 ± 0.182	0.385 ± 0.144	<0.001 (*)	2.460
Apical	1.076 ± 0.412	0.731 ± 0.798	0.295	0.514
Global	1.073 ± 0.228	0.562 ± 0.295	0.002 (*)	1.834

Values are presented as mean ± standard deviation. (*) indicates *p* value < 0.05 for ICI-treated patients compared to controls.

**Table 6 jpm-12-01332-t006:** CMR strain indices of cases vs. controls.

**Controls**	**Mean Longitudinal Systolic Strain (%)**	**Mean Longitudinal Peak Systolic Strain Rate (s^−1^)**	**Mean Longitudinal Early Diastolic Strain Rate (s^−1^)**
1	−16.960	−0.828	0.565
2	−15.825	−0.492	0.661
3	−13.026	−0.619	0.432
4	−17.419	−0.825	0.666
5	−13.159	−0.623	0.352
6	−16.746	−0.748	0.516
7	−13.299	−0.569	0.435
8	−13.941	−0.605	0.412
Average	−15.047	−0.664	0.505
Standard Deviation	1.879	0.123	0.117
**Cases**	**Mean Longitudinal Systolic Strain (%)**	**Mean Longitudinal Peak Systolic Strain Rate** **(s^−1^)**	**Mean Longitudinal Early Diastolic Strain Rate** **(s^−1^)**
1	−18.438	−0.683	0.602
2	−10.935	−0.435	0.253
3	−14.392	−0.633	0.485
4	−11.553	−0.421	0.328
5	−13.515	−0.530	0.338
6	−14.332	−0.656	0.428
7	−11.938	−0.406	0.293
8	−6.908	−0.232	0.094
Average	−12.751	−0.499	0.353
Standard Deviation	3.331	0.155	0.154
*p* value (Controls vs. Cases)	0.112	0.034	0.043

CMR: Cardiac Magnetic Resonance.

**Table 7 jpm-12-01332-t007:** Left ventricular (LV) strain and strain rate characteristics of immune checkpoint inhibitor (ICI)-treated patients compared to the controls using cardiac MRI.

Parameters	Controls (*n* = 8)	ICI-Treated Patients (*n* = 8)	*p* Value	Hedges’s g for Effect Size
LV Systolic Longitudinal Strain (%)				
Basal	−18.359 ± 2.179	−15.725 ± 4.035	0.127	0.768
Midventricular	−14.880 ± 2.752	−12.024 ± 4.278	0.135	0.751
Apical	−10.330 ± 2.846	−9.382 ± 3.540	0.565	0.279
Global	−15.047 ± 1.879	−12.751 ± 3.331	0.112	0.803
LV Peak Systolic Longitudinal Strain Rate (s^−1^)				
Basal	−0.792 ± 0.147	−0.605 ± 0.212	0.060	0.967
Midventricular	−0.652 ± 0.141	−0.491 ± 0.191	0.075	0.909
Apical	−0.490 ± 0.169	−0.353 ± 0.138	0.097	0.840
Global	−0.664 ± 0.123	−0.499 ± 0.155	0.034 (*)	1.113
LV Early Diastolic Longitudinal Strain Rate (s^−1^)				
Basal	0.653 ± 0.162	0.455 ± 0.251	0.082	0.886
Midventricular	0.478 ± 0.159	0.264 ± 0.118	0.008 (*)	1.449
Apical	0.323 ± 0.216	0.333 ± 0.177	0.926	0.045
Global	0.505 ± 0.117	0.353 ± 0.154	0.043 (*)	1.051

Values are presented as mean ± standard deviation. (*) indicates *p* value < 0.05 for ICI-treated patients compared to controls.

**Table 8 jpm-12-01332-t008:** Right ventricular (RV) strain and strain rate characteristics of immune checkpoint inhibitor (ICI)-treated patients compared to the controls using cardiac MRI.

Parameters	Controls (*n* = 8)	ICI-Treated Patients (*n* = 8)	*p* Value	Hedges’s g for Effect Size
RV Systolic Longitudinal Strain (%)				
Free Wall	−19.965 ± 5.617	−13.143 ± 5.168	0.024 (*)	1.195
Septal	−13.794 ± 3.398	−10.186 ± 2.171	0.024 (*)	1.196
Global	−16.879 ± 4.026	−11.665 ± 3.457	0.015 (*)	1.314
RV Peak Systolic Longitudinal Strain Rate (s^−1^)				
Free Wall	−0.861 ± 0.219	−0.636 ± 0.305	0.112	0.801
Septal	−0.573 ± 0.172	−0.401 ± 0.138	0.044 (*)	1.043
Global	−0.717 ± 0.173	−0.518 ± 0.218	0.063	0.954
RV Early Diastolic Longitudinal Strain Rate (s^−1^)				
Free Wall	0.689 ± 0.311	0.381 ± 0.189	0.031 (*)	1.133
Septal	0.357 ± 0.172	0.290 ± 0.095	0.357	0.451
Global	0.523 ± 0.199	0.336 ± 0.108	0.035 (*)	1.105

Values are presented as mean ± standard deviation. (*) indicates *p* value < 0.05 for ICI-treated patients compared to controls.

## Data Availability

The published data will be made available upon satisfactory written request.

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
