# Peer review of "Echocardiographic and Cardiac MRI Comparison of Longitudinal Strain and Strain Rate in Cancer Patients Treated with Immune Checkpoint Inhibitors"

_jpm, 2022, doi:10.3390/jpm12081332_

Round 1

Reviewer 1 Report

This MUST be a prospective study with repeated CMR and echocardiography assessment NOT a study with an independent control group!!!!

If you want to separate the influence of malignant disease per se and previous comorbidities, the study with this aim must be prospective with repeated measurements of cMRI and echocardiographic data during the therapy with checkpoint inhibitors. In this case with so a small number of patients, there are potentially several confounding factors that can influence such sophisticated parameters as strain. So it will be very useful to perform this kind of study with new, potentially cardio-toxic drugs. 

Author Response

We are thankful for the opportunity to present a revised version of our manuscript. The reviewer's recommendations have helped us to strengthen this manuscript.  We have made sincere attempts to address your comments. 

Reviewer 1

Comment 1. This MUST be a prospective study with repeated CMR and echocardiography assessment NOT a study with an independent control group!!!!

If you want to separate the influence of malignant disease per se and previous comorbidities, the study with this aim must be prospective with repeated measurements of cMRI and echocardiographic data during the therapy with checkpoint inhibitors. In this case with so small number of patients, there are potentially several confounding factors that can influence such sophisticated parameters as strain. So, it will be very useful to perform this kind of study with new, potentially cardio-toxic drugs. 

Response: We thank you for this valuable comment. We apologize that we did not clearly explain our study design in the prior version of our manuscript. Briefly, this is a retrospective cohort study studying 8 cases and 16 hospital-based controls. The 8 subjects, however, underwent both CMR and echocardiography within 24-48 hours apart, and were then followed up prospectively. We have made the necessary changes in the manuscript to rectify those issues as described below-

The amended text is provided below (page 1, Abstract)-

Then, we compared mean longitudinal strain and strain rates derived from echocardiography and CMR in the same ICI-treated cohort of patients (n=8). They underwent both of these imaging studies with images taken 24-48 hours apart and followed up prospectively within the same hospital course.

As of your concern regarding the sample size, we measured the Hedges’ g statistic between the different groups which postulated that they were comparable, thus reducing the effect of confounding variables. However, we do recognize the shortcomings of the small sample size and will rectify this in future studies to reduce potential errors.

Reviewer 2 Report

1. The Introduction section is far too long and contains too many parts that should rather be integrated into the Discussion section. Therefore, this section should be shortened significantly, most likely by 1/2 or even by 2/3.

2. Authors should disclose how many CMR/echo operators performed examinations and what was inter-rater and intra-rater variability and kappa coefficients for this? This should be disclosed or discussed as a significant limitation.

3. Discussion should be significantly shortened, it is too dense and difficult to read.

4. What about right ventricular free wall strain? Was this measured?

5. Please add standard deviation for BNP measurement.

Author Response

We are thankful for the opportunity to present a revised version of our manuscript. The reviewer's recommendations have helped us to strengthen this manuscript.  We have made sincere attempts to address your comments. 

Comment 1. The Introduction section is far too long and contains too many parts that should rather be integrated into the Discussion section. Therefore, this section should be shortened significantly, most likely by 1/2 or even by 2/3.

Response We thank the reviewer for this valuable comment. As per your concerns we have made changes to shorten the introduction text as shown below-

The amended text is provided below (page 2, Introduction)-

Advances in cancer treatment have grown to include a new subset of therapy known as immune checkpoint inhibitors (ICIs), which are at the forefront of antineoplastic therapies for several types of cancers. These include anti-cytotoxic T lymphocyte associated antigen CTLA-4 (e.g., ipilimumab, tremelimumab), anti-PD-1 (nivolumab, pembrolizumab, cemiplimab), and anti-PD-L1 (atezolizumab, durvalumab, avelumab). ICI therapy inhibits immune evasion of tumor cells, allowing for T cell-mediated antitumor immunity (1).

While ICI therapies have shown promising results, their toxicity profile includes various autoimmune and inflammatory conditions including cardiotoxicity (2) (3) (4) (5) (6) (7) (8) (9) (10) (11) (12) (13, 14). Cardiotoxicity profile includes systolic heart failure, arrhythmias, pericardial and myocardial disease of which the most common reported is myocarditis when utilized to treat melanoma (15) (16). ICI-induced myocarditis is associated with a high reported mortality rate of 46% (17), with a prevalence of 0.06-2.4% (12) and incidence of about 1%, which doubles with combination therapy (17) (18). Presentation is extremely variable, ranging from asymptomatic cardiac biomarker elevation to life-threatening fulminant myocarditis (19, 20). Screening begins with serial troponin measurements and electrocardiography (ECG), though there are no clear evidence-based guidelines (18) (19, 21, 22).  A 10 ms increment in QRS duration in post ICI myocarditis ECG is associated with increased odds of MACE (23, 24). Non-invasive imaging modalities such as echocardiogram and cardiac magnetic resonance (CMR) allow for better tissue characterization. Endomyocardial biopsy (EMB) is the gold standard for confirmation, however at least six biopsies from different regions are needed given the patchy T-cell myocardial infiltration, even so there is a high false negative rate (12). A retrospective study conducted CMR strain analysis, indicating that despite normal left ventricular function, strain imaging was abnormal, thus highlighting its importance (25).

In this study, we aim to further study the utility of CMR in comparison to echocardiography in patients suspected to have developed ICI related adverse events. In particular, we focused on the regional and global comparisons mean longitudinal systolic strain, peak systolic strain rate and early diastolic strain rate between these modalities.

Comment 2. Authors should disclose how many CMR/echo operators performed examinations and what was inter-rater and intra-rater variability and kappa coefficients for this? This should be disclosed or discussed as a significant limitation.

Response Cardiac MRI images were analyzed by U.C.S (a level 3 certified MRI expert in our institution) and S.S.S. (a physician).  Echocardiography images were analyzed by B.K. (4 years of experience with MRI post processing) and S.S.S. (a physician). We have also added additional text referring inter-rater variability and kappa co-efficients.

The amended text is provided below (page 4, 2.6. Statistical Analyses)-

The interrater reliability was measured as per Cohen’s kappa statistic. B.K. and S.S.S. analyzed echocardiography findings with a kappa value of 0.500 (95% CI -0.020 to 1.0000) suggestive of moderate agreement. U.C.S. and S.S.S. analyzed CMR and had a kappa value of 1.0000 (95% CI 1.0000 to 1.0000) reported an almost perfect agreement. These values were calculated using Graph Pad Prism 9.1.2 (La Jolla, CA) software.

Inter-modality agreement between the quantitative measurements of mean longitudinal strain and strain rates calculated through echocardiography and CMR were presented with Bland-Altman plot and correlation analysis. Unfortunately, we were not able to perform the Intra-rater reliability due to the relatively small sample size and a risk of operator recall bias.

Comment 3. Discussion should be significantly shortened; it is too dense and difficult to read.

Response We have adequately shortened the discussion.

The amended text is provided  in the discussion section (page 14, 4. Discussion).

Comment 4. What about right ventricular free wall strain? Was this measured?

Response As per your recommendation we have added the effects of ICI on right ventricular free wall strain and values are listed below-

The amended text is provided below (page 11, 3.7. Cardiac MRI Characteristics)-

Global

0.505 ± 0.117

0.353 ± 0.154

0.043 (*)

1.051

Values are presented as mean ± standard deviation. (*) indicates p value < 0.05 for ICI-treated patients compared to controls.

Table 8. Right ventricular (RV) strain and strain rate characteristics of immune checkpoint inhibitor (ICI)-treated patients compared to controls using cardiac MRI.

Parameters

Controls (N=8)

ICI-Treated Patients (N=8)

p value

Hedges’s g for Effect Size

RV Systolic Longitudinal Strain (%)

Free Wall

-19.965 ± 5.617

-13.143 ± 5.168

0.024 (*)

1.195

Septal

-13.794 ± 3.398

-10.186 ± 2.171

0.024 (*)

1.196

Global

-16.879 ± 4.026

-11.665 ± 3.457

0.015 (*)

1.314

RV Peak Systolic Longitudinal Strain Rate (s-1)

Free Wall

-0.861 ± 0.219

-0.636 ± 0.305

0.112

0.801

Septal

-0.573 ± 0.172

-0.401 ± 0.138

0.044 (*)

1.043

Global

-0.717 ± 0.173

-0.518 ± 0.218

0.063

0.954

RV Early Diastolic Longitudinal Strain Rate (s-1)

Free Wall

0.689 ± 0.311

0.381 ± 0.189

0.031 (*)

1.133

Septal

0.357 ± 0.172

0.290 ± 0.095

0.357

0.451

Global

0.523 ± 0.199

0.336 ± 0.108

0.035 (*)

1.105

Values are presented as mean ± standard deviation. (*) indicates p value < 0.05 for ICI-treated patients compared to controls.

Page 10, 3.7. Cardiac MRI Characteristics

As illustrated in Table 8, ICI-treated patients had significantly reduced right ventricular (RV) longitudinal strains both globally as well as in the free wall and septal regions. Compared to controls, ICI-treated patients experienced significant reductions in RV SRs in the septal region and RV SRE in the free wall region. These results indicated that ICI-associated cardiotoxicity might have an adverse impact on RV contractility.

Comment 5. Please add standard deviation for BNP measurement.

Response We have added standard deviation of BNP as listed below-

The amended text is provided below (page 5, Table 1. Demographic characteristics of patients, cancer type, immune checkpoint inhibitor (ICI) treatment, cardiac biomarkers)-

Mean Maximum BNP, ng/mL

136 (Range 34-318 pg/mL)

We again appreciate the reviewer’s constructive criticism and we believe that the revisions to the text have substantially improved the manuscript.  

Round 2

Reviewer 1 Report

Your investigation is an essential beginning of the long journey. More extensive study with better designs is needed. I have no further comments.

You pull out the maximum from your data and you are in the right direction. 

Author Response

Thank you!

Reviewer 2 Report

Thank you for attending to all of my comments efficiently.

I have no further questions.

Author Response

Thank you!